# Compact all-fiber quantum-inspired LiDAR with over 100 dB noise rejection and single photon sensitivity

Han Liu [1] ✉, Changhao Qin [1], Georgios Papangelakis [1], Meng Lon Iu[1] & Amr S. Helmy[1]

Entanglement and correlation of quantum light can enhance LiDAR sensitivity in the presence of strong background noise. However, the power of such quantum sources is fundamentally limited to a stream of single photons and cannot compete with the detection range of high-power classical LiDAR transmitters. To circumvent this, we develop and demonstrate a quantum-inspired LiDAR prototype based on coherent measurement of classical time-frequency correlation. This system uses a high-power classical source and maintains the high noise rejection advantage of quantum LiDARs. In particular, we show that it can achieve over 100dB rejection (with 100ms integration time) of indistinguishable (with statistically identical properties in every degree of freedom) in-band noise while still being sensitive to single photon signals. In addition to the LiDAR demonstration, we also discuss the potential of the proposed LiDAR receiver for quantum information applications. In particular, we propose the chaotic quantum frequency conversion technique for coherent manipulation of high dimensional quantum states of light. It is shown that this technique can provide improved performance in terms of selectivity and efficiency as compared to pulse-based quantum frequency conversion.

In any optical sensing instrumentation, the light source and detection system used play a pivotal role in dictating the performance. In recent years, a radical approach to enhancing optical sensing system sensitivity has been to use quantum light sources and measure their non-classical properties. This serves to surpass the performance limit imposed by the classical laws of physics. A manifestation of this idea in the target detection domain is quantum illumination (QI), where quantum entanglement is utilized to reject the background noise of the target detection channel[1–4]. In a QI setup, the probe light that is entangled with, the locally stored reference light, interrogates the target. Back-reflected probe light (if the target is present) mixed with strong background noise light is collected and undergoes a joint detection measurement along with the reference light to determine the target's presence or absence. In contrast to the common perception of quantum light being

fragile, the performance advantage of QI over classical detection is most pronounced in the high loss and high noise regime. Similar enhancement of LiDAR sensitivity has also been demonstrated through phase-insensitive measurement of photon-photon correlations[5–7].

Despite its unrivaled performance over classical LiDARs with equal probe power, practical applications of QI are severely curtailed owing, in part to its fundamental power limit: not only is the flux of an entangled light source difficult to increase, but also the performance enhancement it offers diminishes as the power increases[8–10]. As such, it is difficult for QI to meet the demand of real-world sensing applications where high probe power is needed to extend the detection range beyond that of a laboratory setup. Therefore, a natural line of inquiry could pose the question of whether it is possible to borrow methodology from QI to enhance classical LiDAR protocols while retaining

[1]The Edward S. Rogers Department of Electrical and Computer Engineering, University of Toronto, 10 King's College Road, Toronto, ON M5S 3G4, Canada. ✉e-mail: qwerty.liu@mail.utoronto.ca

the essential performance enhancement. Similar approaches, other than QI have already been proven successful for sensing protocols that were initially believed to rely on quantum effects, those include ghost imaging[11], and quantum optical coherence tomography[12,13].

In this work, we demonstrate a LiDAR design and the associated prototype with a similar setup and operation principle as a QI, except that it uses a classical time-frequency correlation source with high power thereby greatly enhancing the potential for achieving a significant operating range. It is shown that by using classical time-frequency correlation, probe light that has random and chaotic time-frequency characteristics can be selectively converted to a single frequency with near-unity quantum efficiency while in-band noise with identical time-frequency characteristics can be reduced to a negligible level. In addition, it is interesting to note that this noise rejection technique is conceptually related to quantum frequency conversion (QFC)[14–18], a quantum information processing technique in which a particular time-frequency mode (corresponding to LiDAR probe light) is selectively separated from the rest of the band (corresponding to indistinguishable noise) with preserved quantum properties. For this reason, we term the LiDAR protocol as chaotic-QFC LiDAR. Theoretical analysis shows that the high-dimensional nature of chaotic-QFC can provide substantially improved performance in terms of efficiency and selectivity as compared to conventional Hermite-Gaussian mode based QFC.

## Results and discussion
### Theoretical modeling of chaotic-QFC LiDAR
The probe and reference light of chaotic-QFC LiDAR is generated through difference frequency generation (DFG) between a single frequency pump ($2\omega_0$) and broadband amplified spontaneous emission (ASE, frequency higher than $\omega_0$). In the limit of broadband DFG phase matching, the probe and reference light temporal amplitudes $A_p, A_r$ (with carrier frequency $\omega_0$ subtracted) are chaotic and complex conjugated to each other. Under the quasi-cw assumption, the probe and reference light can be modeled as stationary Gaussian random processes that are characterized by their correlation function and the phase conjugation relation:

$$\langle A_p(t)A_p^*(t')\rangle = P_p f(t - t') \quad A_r(t) = \sqrt{\frac{P_r}{P_p}}A_p^*(t) \tag{1}$$

where $P_p, P_r$ are the photon flux of the probe and reference light, respectively, and angled brackets stand for statistical averaging. For simplicity, the correlation function $f$ is assumed to be Gaussian with variance $1/\sigma^2$ ($\sigma$ is the common bandwidth of probe and reference light). The chaotic and conjugated phases of the probe and reference light can be understood as classical time-frequency correlation as will be discussed later. Background noise light with flux $P_n$ will naturally be uncorrelated with the reference light and is assumed to have the same time-frequency characteristics as the probe light. This condition is a worst-case scenario asymptotic case, where it is assumed that the noise involved is fully in-band and cannot be removed through simple filtering. In the LiDAR transceiver, probe light is mixed with strong background noise and collected by a receiving telescope.

At the receiver section, the locally stored reference light can selectively convert probe light to another frequency via SFG, with little crosstalk from the noise light that has identical time–frequency properties. To analyze this, consider first the small signal SFG regime, in which the SFG output amplitude is given by (see Supplementary Note 2):

$$A_{SFG}(t) = \frac{\gamma}{\Delta\beta}\left(A_p(t)A_r(t) + A_n(t)A_r(t)\right) * \Pi\left(\frac{t}{\Delta\beta L}\right) \tag{2}$$

where $\gamma, L$ specifies the normalized nonlinearity and waveguide length and * stands for convolution. The rectangle function $\Pi(t)$ equals unity

for $-1/2 < t \le 1/2$ and zero otherwise. The inverse group velocity difference $\Delta\beta$ is defined as the difference between the inverse group velocity of SFG and probe light. The group velocity of noise and reference light is assumed to be the same as the probe light if the dispersion around $\omega_0$ is negligible[15]. Because the probe, reference, and noise light are all stationary random processes, so is the SFG output. Therefore the SFG power spectral density $S(\omega)$ is given by the Fourier transform of the autocorrelation of $A_{SFG}(t)$:

$$S(\omega) = \gamma^2 L^2 \text{sinc}^2\left(\frac{\Delta\beta L\omega}{2}\right)\left\{P_p P_r \delta(\omega) + \frac{(P_n + P_p/2)P_r}{2\sqrt{\pi}\sigma}\exp\left(-\frac{\omega^2}{4\sigma^2}\right)\right\} \tag{3}$$

where $\delta(\omega)$ is the delta function. The two terms above can be understood as follow: the random, but correlated phases of the reference and probe light cancel each other and result in a single-frequency coherent SFG (c-SFG) peak. In contrast, the noise and reference light will only contribute to broadband incoherent SFG (i-SFG). It is worth noting that the probe and reference light will also generate a small amount of i-SFG (as compared to c-SFG) in the absence of noise light. This is because the inherent intensity fluctuation of the chaotic probe and reference light will create a chaotic component (i-SFG) of the SFG output. Noise reduction of chaotic-QFC LiDAR is achieved by applying narrowband (optical and electrical) filtering to separate c-SFG from i-SFG. The signal-to-noise ratio (SNR) of chaotic-QFC LiDAR is then given by the ratio between c-SFG and i-SFG power:

$$\text{SNR}_{QFC} = \frac{2\sqrt{\pi}\sigma}{BW}\frac{2P_p}{2P_n + P_p} \tag{4}$$

where $BW$ is the filter bandwidth. If the receiver does not have access to the correlation information (reference light), background noise will appear to be completely indistinguishable from the probe light. Then the only useful information that can be extracted from collected light is the change of optical power due to target reflection. The SNR of such direct detection is given by $\text{SNR}_{DD} = P_p/(P_n + P_p)$, compared to which the SNR enhancement of chaotic-QFC LiDAR is given by:

$$\frac{SNR_{QFC}}{SNR_{DD}} = \frac{2\sqrt{\pi}\sigma}{BW} \tag{5}$$

For example, if the probe spectrum is 7.5 nm wide around 1560 nm (in full-width half max, obtainable with commercial filters for fiber optical communication) and the filter bandwidth is 10 Hz, the SNR enhancement is around 111 dB.

The chaotic-QFC LiDAR sensitivity is dictated by the c-SFG efficiency beyond the small signal SFG regime, which is defined as the probability of converting an incoming probe photon to a c-SFG photon. To maximize the c-SFG efficiency, the i-SFG power needs to be minimized due to the photon number conservation constraint. This can be achieved with a narrow SFG phase matching bandwidth (long waveguide) or cavity resonance at the c-SFG frequency[19]. In the limit of negligible i-SFG power, the c-SFG efficiency can be analytically solved beyond the small signal SFG regime(see Supplementary Note 2):

$$\eta_{\text{cSFG, }narrowband} = \sin^2(\gamma\sqrt{P_r}L)\exp(-\beta^2\Delta L^2\sigma^2) \tag{6}$$

where $\Delta L$ is the relative distance between the probe and reference light. In particular, for zero relative delay $\Delta L = 0$ and some finite reference power $P_r$, the coherent conversion efficiency can reach 100%, provided that the dispersion of probe and reference light (due to freespace and fiber propagation) is properly compensated (see Supplementary Note 2). Such conversion will also preserve the quantum properties of the input probe light because c-SFG is an intrinsically

noiseless process that acts like a frequency domain beam-splitter[20]. In the general case of non-negligible i-SFG power, the c-SFG efficiency can be calculated through Monte Carlo simulation by drawing different samples of the chaotic probe and reference amplitudes and numerically solving the coupled mode equations (see Supplementary Note 2). Numerical results show that for a 5cm long waveguide with 0.09 group index difference (inferred from a second harmonic generation characterization) between SFG and probe/reference light, the maximal c-SFG efficiency can reach 92%. Unlike QI, the target distance for chaotic-QFC LiDAR does not need to be stabilized down to the sub-wavelength level even though a coherent receiver is used. This is because the c-SFG power is not fast varying as a function of the probe light phase. The distance of the target, however, can be determined by scanning the reference light delay and monitoring the c-SFG power. The distance resolution is inversely proportional to the probe light bandwidth ($\simeq$100 μm for 7.5 nm probe bandwidth).

The schematic of different stages of the chaotic-QFC LiDAR system and corresponding spectra of interacting light waves are summarized in Fig. 1a–g. The probe light is created as a spectral mirror image (Fig. 1c) of the reference light (ASE, Fig. 1b) with conjugated phase, as per the energy conservation requirement of DFG. The probe light is mixed with noise in the freespace transceiver (Fig. 1d). Before SFG, the group velocity dispersion and relative delay of the probe and reference light need to be compensated, such that different frequency components of the probe and reference light contribute to c-SFG constructively. The filtering of i-SFG consists of three stages: (1) the phase matching bandwidth (Fig. 1e) limits the i-SFG generation spectrum and (2) an optical bandpass filter (Fig. 1f) to optically reject most i-SFG power and (3) a narrowband electrical filter (integration time) of the phase-sensitive detection (balanced homodyne) signal (Fig. 1g) to maximize the noise rejection. To avoid low-frequency technical noise, the frequency of the homodyne detection signal can be shifted to a non-dc frequency $f_{HDD}$ through sawtooth wave modulating (serrodyne) the probe light(Fig. 1g).

## Experimental Results

The chaotic-QFC LiDAR experimental setup is shown in Fig. 2 and its details can be found in the Method section. In the source section, the correlated probe and reference light is generated through DFG between single frequency (781.7 nm) pump light and ASE light. The generated probe and reference light is separated and boosted up in power with spectral filters and fiber amplifiers. The relative delay and total dispersion of the probe and reference arm are compensated for maximal c-SFG efficiency. The transceiver section consists of a probe light collimator, a homemade telescope, and a target object placed around 4 m away from both the collimator and the telescope. Two different target objects are tested: a piece of grounded glass stabilized on the optical table and a quadcopter drone (with rough plastic surfaces) placed on top of a cart outside the optical table. At the output of the transceiver, the probe light is mixed with noise light that has the same time-frequency statistical properties. In the receiver section, collected probe light and reference light undergo SFG in a periodically poled lithium niobate (PPLN) waveguide (HCP-RPE-5 cm, details in coating, loss, and phase matching information can be found in Supplementary Note 3). The output of the SFG first goes through a narrowband optical filter (diffraction grating) and then is phase-sensitively detected through balanced homodyne detection. The optical spectra of all interacting waves are shown in Fig. 3a.

The LiDAR sensitivity is benchmarked by the SFG and c-SFG efficiency of probe photons (see the Method section for details). Experimental results are in agreement with Monte Carlo simulation (Fig. 3b). As can be seen, the c-SFG efficiency with around 260 mW input reference power reaches up to 90%. The measurement uncertainty is mainly due to the instability of the DFG pump laser's coherence.

The LiDAR noise resilience is benchmarked by comparing the homodyne signal level for different probe and noise powers (as measured at the output of the SFG waveguide). It is worth emphasizing that the noise light has identical time-frequency properties as probe light and therefore cannot be separated from probe light via conventional time-frequency filtering. Instead, the probe light is converted to c-SFG and separated from noise light-induced i-SFG via narrowband optical and electrical filtering. In the current experimental setup, the waveguide phase matching and diffraction grating contribute to around 97% reduction of i-SFG (0.12 nm filter bandwidth being applied on 3.7 nm non-phase matched i-SFG bandwidth). Additional optical filtering can be applied by adding a high-fineness Fabre-Perot filter.

The narrowband electrical filtering is simulated by using 10Hz resolution bandwidth (100ms integration time) of an electrical spectrum analyzer (ESA). As can be seen in Fig. 4, to produce the same homodyne signal level, the noise power needs to be 107dB higher than probe light back-reflected from a stabilized target. The reason for the measured 107 dB noise rejection being lower than the theoretical prediction (111dB from Eq. (5)) can be attributed to many factors, including nonideal serrodyne modulation, non-Gaussian shaped

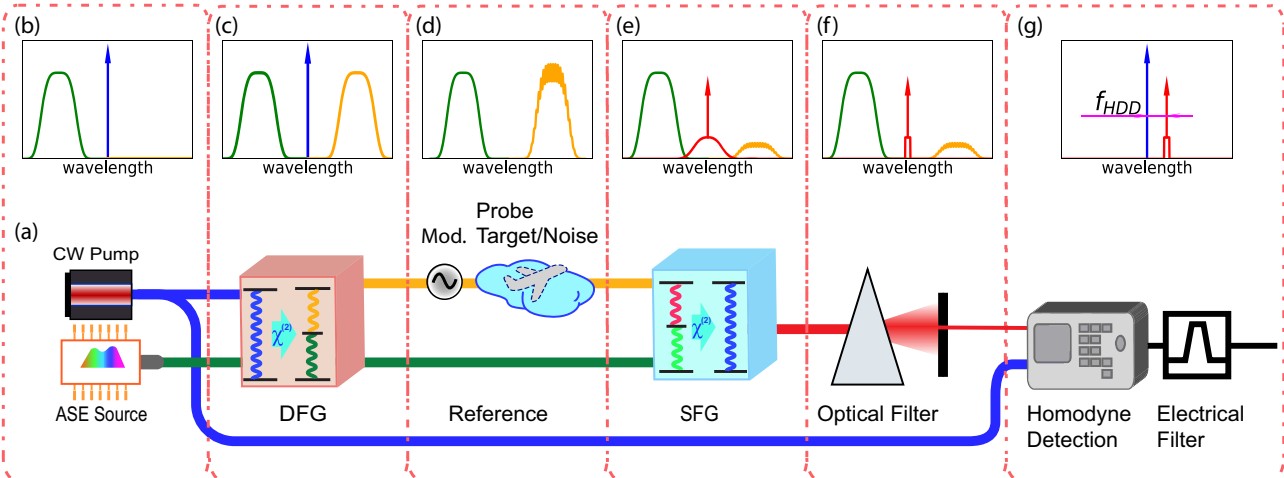

**Fig. 1 | The schematic diagram of the chaotic-QFC LiDAR and its various stages. a** Overview, **b** the spectrum of the DFG pump (blue) light and ASE (green) before the DFG process, **c** the spectrum of the probe (yellow), reference (green), and DFG pump light after the DFG process, **d** the spectrum of the reference and collected probe light back-scattering (mixed with noise), **e** the spectrum of the reference light, depleted probe light, c-SFG peak (red arrow) and unfiltered i-SFG (red curve) background, (**f**) the narrowband optical filtering of i-SFG, **g** the homodyne beating frequency $f_{HDD}$ between the DFG pump light and frequency shifted c-SFG.

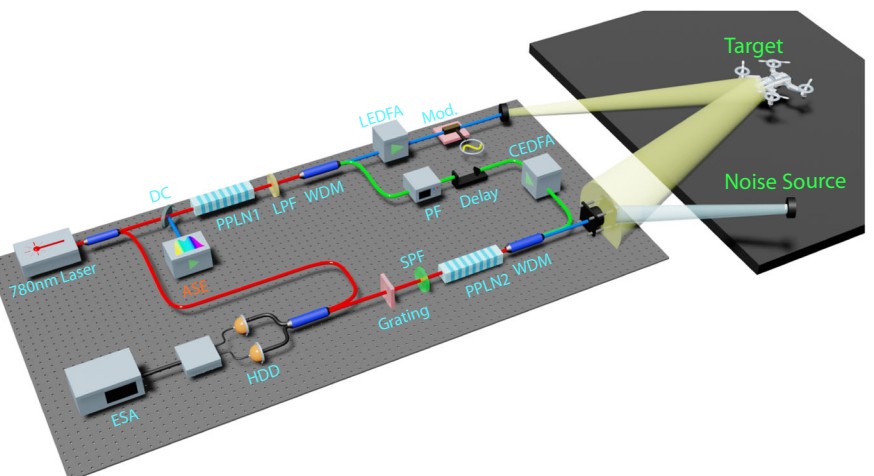

**Fig. 2 | The experimental setup of the chaotic-QFC LiDAR system.** DC dichroic combiner, WDM wavelength demultiplexer, Mod phase modulation for frequency shifting (serrodyne), PF programmable amplitude-phase filter, HDD balanced homodyne detection, ESA electric spectrum analyzer, SPF/LPF short/long pass filter, ASE amplified spontaneous emission, PPLN1, 2 periodically poled lithium niobate waveguide for DFG and SFG, Target: quadcopter drone on a separate table, CEDFA,LEDFA:C and L band erbium doped fiber amplifiers.

spectra of probe and reference light, etc. When the unstabilized quadcopter is used as the target object, back-reflected probe light (with the same flux as the stabilized target case) generated a weaker homodyne signal and provides less noise rejection (95 dB). This is because mechanical vibrations induce $\simeq 1$ kHz Dopler broadening of the back-reflected probe light. Such phase noise is transferred to the c-SFG spectrum (Fig. 5). As a result, the 10 Hz filter also reduces the level of the measured homodyne signal due to over-filtering (10 Hz < 1000 Hz). It is also worth noting that in the current setup, the nonzero DFG pump bandwidth (75 kHz) does not spectrally broaden the homodyne signal. This is because phase fluctuations of c-SFG and the homodyne local oscillator light are identical and cancel each other.

To benchmark the ranging performance, the delay of the reference light is scanned with a homemade mechanical delay line to determine the target object(quadcopter) distance (Fig. 6). When scanning, a Dopler frequency shift of homodyne signal that is proportional to the scanning speed is observed. The homodyne signal level as a function of the scan distance shows multiple side peaks, which is possibly caused by diffusive reflections of probe photons into different directions, resulting in different probe path lengths inside the telescope. It is confirmed that when an object with specular reflection (mirror) is used as the target object, the ranging signal is a single peak localized in distance. It is also worth mentioning that the speed and dynamic range of delay scanning in our current setup can potentially be improved with commercial solid-state optical switches, and fixed fiber optical delay lines(see Supplementary Note 4).

### Classical and quantum time–frequency correlation
The idea of utilizing sum frequency generation to analyze classical time-frequency correlation has been demonstrated in chirped pulse interferometry (CPI)[12,13]. However, unlike in CPI where correlation is manually created by preparing laser pulses with opposite chirps, the correlation used in chaotic-QFC LiDAR originates from the random and chaotic nature of broadband ASE light and the DFG energy conservation constraint, in a way that is similar to the generation of time-frequency entangled photon pairs. This can be seen from Eqs. (3) and (6) as follows: probe and reference light generate SFG mostly at a single frequency (c-SFG) despite both being broadband in the spectrum. Also, c-SFG is efficient only if the probe and reference light has zero relative delay, despite both being stationary in time. This is similar to the SFG process of time-frequency entangled photon pairs where

two photons must have simultaneous time-correlation and frequency anti-correlation to produce the SFG photon at a specific frequency[4,21]. Also, time-frequency entanglement and classical correlation are both generated via parametric down-conversion processes but with different input stimulation (vacuum versus ASE). The essential difference between the classical correlation and entanglement is that: while entanglement can be simultaneously measured in time and frequency domain through joint measurement[4], classical time and frequency correlation are actually the same correlation being represented in the time or frequency basis. As a result, classical time-frequency correlations are subjected to the time-frequency uncertainty constraint. For LiDAR, this implies that the noise rejection provided by classical correlation is exactly half (in log scale) of that provided by entanglement. Nevertheless, the classical nature of the correlation allows for probe and reference power beyond the single photon level and the option of classical amplification. This is of paramount utility for practical applications where high probing power is required to extend the detection range beyond that of a proof-of-principle laboratory setup with single photon sources.

### Comparison to other LiDAR protocols
The noise light for chaotic-QFC LiDAR is assumed to be statistically identical to probe light since out-of-band noise (in either time or frequency domain) can be filtered without impacting the LiDAR performance. Also, from a practical point of view, indistinguishable noise light simulates active jamming attacks or unintentional interference (e.g. identical adjacent LiDARs operating at the same time), which are important safety concerns that degrade the sensitivity and increase the error rates of LiDARs[22]. The conventional approach to this problem is to generate spread spectrum probe light and perform correlation analysis in the rf domain, i.e. chaotic LiDARs based on chaotic lasers[23] and random modulation[22,24]. In comparison, the time-frequency correlation used in chaotic-QFC LiDAR can be considered a spread spectrum technique in the optical domain with two vital differences. First, the optical time-frequency correlation is much broader in bandwidth than the rf domain spread spectrum. This translates to orders of magnitude superior noise suppression and ranging resolution. Second, the time-frequency correlation is optically analyzed through optical nonlinear processes, and doing so avoids fundamental limitations of electrical domain measurement and signal processing such as detector saturation, electrical interference, finite sampling rate, digitization noise, and imperfect common mode rejection ratio of

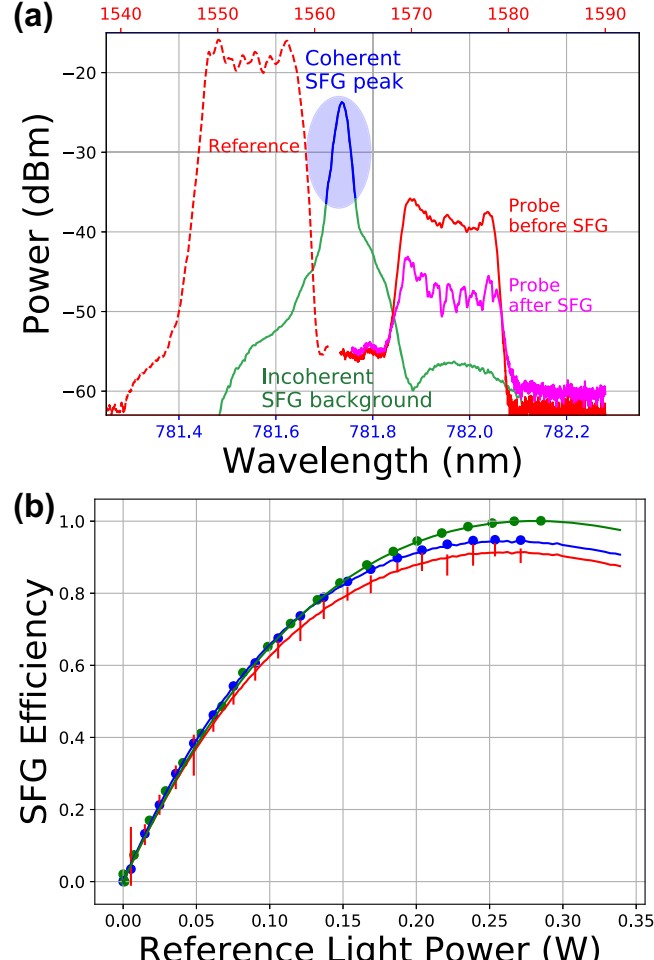

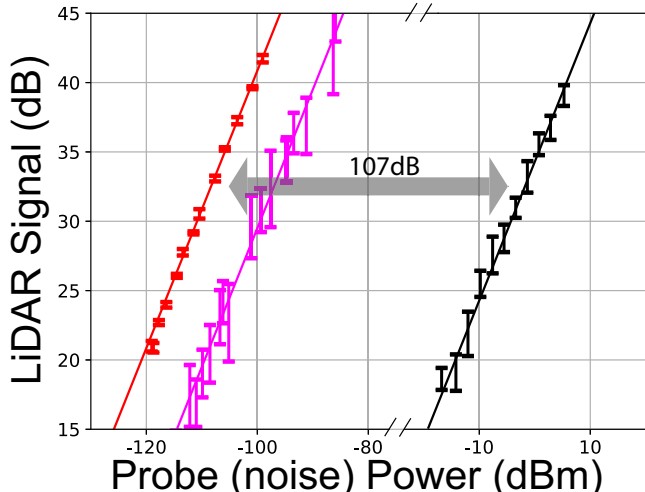

**Fig. 4 | The homodyne signal level for different probe and noise power.** Red: probe light back-reflected from a rough glass surface mounted on the optical table, magenta: probe light back-reflected from the quadcopter with phase noise induced by mechanical vibration, black: noise light. The reference power is kept around 260 mW to maximize the conversion efficiency. The signal levels are measured relative to the shot noise level, with 100 ms integration time.

**Fig. 3 | The SFG waveguide output spectra and conversion efficiencies. a** the spectrum of reference light (dashed red), probe light without SFG (solid red), probe light with SFG (magenta), and SFG light (blue). The SFG spectrum (0.01 nm resolution) consists of a c-SFG peak and broadband i-SFG background. The absolute level of the SFG spectrum is not shown. **b** the theoretically predicted (curve) and experimentally measured (errorbar or dot) c-SFG efficiency (red) and total SFG efficiency (blue). The SFG efficiency of the single-frequency probe and reference light is also plotted (green dot and curve) as a baseline. The error bars are obtained by taking the standard deviation of three repeated measurements.

balanced photodetection, etc. A brief review and comparison of different types of LiDAR based on classical correlation are given in Supplementary Note 1[22–26].

## Chaotic-QFC for quantum information processing

In chaotic-QFC LiDAR, probe light with a specific chaotic amplitude is separated from uncorrelated noise light via c-SFG. It is interesting to note that a one-to-one correspondence can be made between this and the process of QFC: in QFC, a quantum state of light (corresponding to the LiDAR probe light) in a particular time-frequency mode (probe mode) is coherently converted to another mode (the c-SFG mode) and separated from other quantum states (noise light), through nonlinear interaction with a strong pump light (reference light). Also, the high-efficiency nature of chaotic-QFC preserves the quantum properties[14] of the input probe light like other QFC protocols. For this reason, chaotic-QFC can be considered as a variant of QFC that is equipped with chaotic modes (CMs) defined by chaotic temporal amplitudes. Nevertheless, there are important differences between chaotic-QFC from conventional QFC based on Hermite-Gaussian modes, in terms of orthogonality and statistical equivalence. These two differences have

important implications on QFC performance metrics, which are summarized as follows.

First, different CMs are approximately orthogonal due to their independent chaotic nature. Nevertheless, if each CM has a sufficiently large time-bandwidth product, such approximate orthogonality is close to exact and can simultaneously provide high conversion efficiency and selectivity (the conversion efficiency ratio of the target mode versus other crosstalk modes[16,27]). To give a concrete example, consider a target CM with 100ns duration and 1THz bandwidth. Since the 100 ns pulse width is much longer than the nonlinear interaction time, the previous quasi-cw modeling of chaotic-QFC still applies. Therefore the c-SFG efficiency will still reach over 90% as has been experimentally observed (can be further improved with narrower SFG phase matching). The generated c-SFG is a 100 ns pulse with a single carrier frequency, whereas i-SFG from other crosstalk modes is evenly distributed around 1THz bandwidth. As a result, by using a $\simeq 10$ MHz optical filter, the i-SFG power can be rejected by around $\simeq 1-10^{-5}$. In comparison to this, state of art QFC based on cascaded nonlinear conversion has been shown to provide $\simeq 90\%$ of crosstalk mode rejection[27].

Another important difference of CMs is that they have similar "noise-like" time-frequency characteristics, whereas Hermite Gaussian modes of different orders have substantially different time-frequency amplitudes. Therefore it is much easier for chaotic-QFC to multiplex many CMs in parallel to span a large Hilbert space for high-dimensional quantum information processing. This can be carried out, for example, by taking different temporal segments of an ASE source, which can be done with integrated optical delay lines. An additional advantage of chaotic-QFC as compared to Hermite-Gaussian mode based QFC is that the generated SFG is in the same time-frequency mode (c-SFG) regardless of the input CMs. This makes the subsequent quantum processing of SFG light easier to standardize. Moreover, this also allows for the interference of c-SFG from different probe modes. When combined with the recently developed multiple output QFC technique[28–30], the chaotic-QFC technique can be useful for frequency domain Boson sampling[31]. The superior performance of chaotic-QFC does, however, come at a price of reduced channel capacity: to ensure approximate orthogonality, the number of CMs must be much lower than the total time-bandwidth product of the channel to ensure sufficiently low mutual overlap.

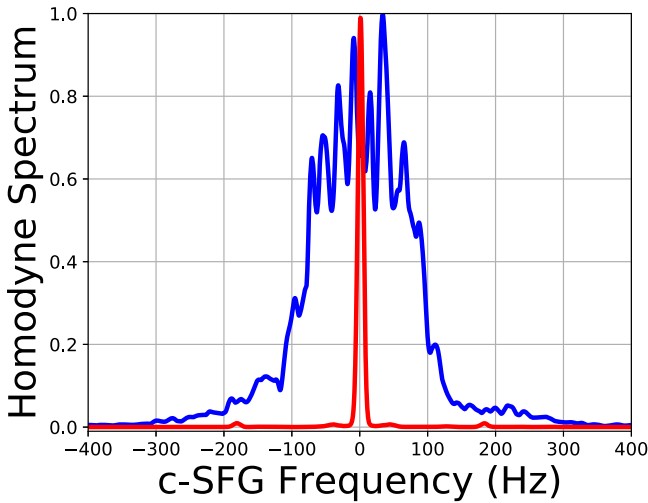

**Fig. 5 | The ESA spectra of the homodyne signal.** Red and blue curves correspond to when the stabilized ground glass or quadcopter is used as the target. Both spectra are normalized in the maximal spectral power level.

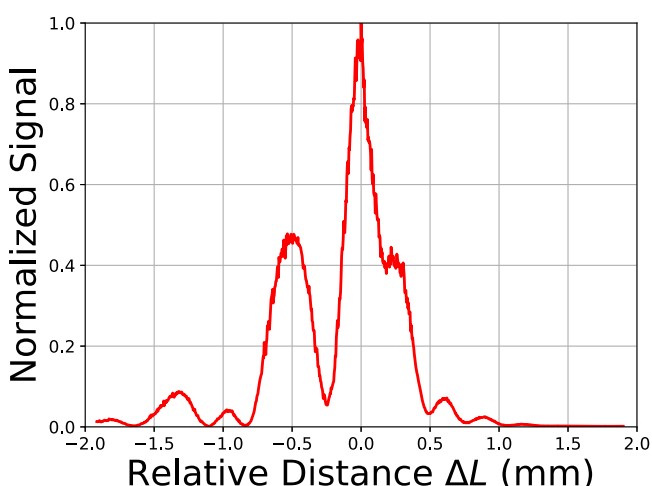

**Fig. 6 | The normalized homodyne signal as a function of the reference light (relative) delay.** The resolution bandwidth is increased to 1 kHz to reduce the impact of the Dopler effect on the homodyne signal level.

It is worth emphasizing that chaotic-QFC, like other QFC protocols, does not generate quantum states of light. Instead, it serves as a means to manipulate quantum light that already exists in CMs through SFG. Therefore, the question remaining is whether quantum states of light can be generated in CMs by some other nonlinear processes in the first place. After all, a quantum light generating process such as spontaneous parametric down-conversion (SPDC), does favor an intrinsic set of modes (e.g. Schmidt modes[32] and Bloch-Messiah modes[33]) depending on the nonlinear medium and pumping. Nevertheless, it is interesting to note that a phase-conjugated pair of CMs (like the probe and reference mode defined by the DFG outputs) can be considered a "approximately" intrinsic (Schmidt) mode pair of SPDC processes that have single frequency pump and broadband phasematching. For such SPDC processes, the joint temporal amplitude (JTA) of generated photon pairs can be approximated by a delta function $\delta(t_p - t_r)$, whose Schmidt decomposition is not unique: an arbitrary pair of probe and reference mode whose temporal amplitude are conjugate to each other can be considered as

a term of the Schmidt decomposition. Such lack of uniqueness implies if a photon is created in a CM mode through the SPDC process, its twin photon will also be created in the conjugated CM that has complex conjugated temporal amplitude. A similar argument can also be generalized for realistic photon pair states whose JTAs are normalizable.

In summary, we proposed a quantum-inspired LiDAR prototype based on coherent measurement of classical time-frequency correlation. It retains the high noise resilience advantage (>100 dB rejection of indistinguishable noise) of quantum LiDARs while still allowing for high-power classical sources and single photon sensitivity to extend the detection range. Its principle also resembles chaotic LiDAR but the implementation with coherent optical processing avoids fundamental limitations of electrical domain detection and signal processing. As such, the LiDAR prototype we demonstrate here combines both practical implementation and substantial performance enhancement. We expected it to become a useful tool in the near future for real-world LiDAR application that requires high rejection of crosstalk and noise jamming. The chaotic mode conversion technique that is derived from the LiDAR receiver, can also be applied in quantum information applications and provide performance enhancement as compared to pulse-based quantum frequency conversion. Its advantage of high efficiency and selectivity can be useful for high dimensional quantum information processing applications such as Boson sampling.

## Method
### The experimental setup
To generate correlated probe and reference light, ASE light from an erbium doped fiber amplifier is used as the input of the DFG process. The bandwidth of the ASE is limited to around 7.5 nm by a tunable band pass filter (Alnair labs BVF-200). A wavelength demultiplexing filter and erbium-doped fiber amplifiers in C and L band are then used to separate and boost up the probe and reference light power. The compensation of the relative delay and total group velocity dispersion (1.6 ps/nm) of the probe and reference arm is done with a programmable amplitude-phase filter (Finisar waveshaper 1000A) and a tunable optical delay line. An adjustable level of noise light is obtained by tapping the probe amplifier output. Since the tapped probe amplifier output is not correlated with reference light due to unbalanced delay, it can be regarded as uncorrelated background noise that has the same time-frequency properties as the probe light. To conduct homodyne measurement of the SFG light at a non-dc frequency, the probe light is frequency-shifted by 2.1 MHz through serrodyne modulation. The remaining non-converted probe and reference light at the SFG output is spectrally separated and the power is monitored.

### Calculation of the SFG efficiencies
The total SFG efficiency is determined from the depletion of probe power throughput when the reference light is turned on and off. The c-SFG efficiency is determined from the ESA signal level at the serrodyne modulation frequency (2.1 MHz), up to a proportional constant that is dictated by many experimental factors including optical, detection, and modulation efficiencies. To calibrate the measured c-SFG efficiency to be independent of this proportional constant, a baseline measurement (Fig. 3b, green dots) is done with single-frequency probe and reference light. Such single-frequency SFG processes can only produce c-SFG without i-SFG due to the energy conservation constraint.

## Data availability
Data sets generated during the current study are available from the corresponding author on request.

## Code availability

The algorithm for the Monte-Carlo SFG simulation is described in the supplementary information and the code is available upon request.

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

## Acknowledgements

The project has been supported by NSERC an IDEaS programs.

## Author contributions

H.L. conceived the idea. H.L. and M.L.I. designed the experiment. C.Q., H.L., G.P. conducted the experiment. H.L. and A.H. wrote the manuscript. All authors discussed the result and commented on the manuscript.

## Competing interests

The authors declare no competing interests.
