## [Peer Review File · Nature Communications]

Compact All-Fiber Quantum-Inspired LiDAR with Over 100dB Noise Rejection and Single Photon SensitivityREVIEWER COMMENTS

Reviewer #1 (Remarks to the Author):

The authors present their work on a new LiDAR scheme inspired by other LiDAR methods that use quantum illumination. The results show a high level of noise rejection without relying on entangled photon pairs thereby allowing ranging over longer distances.

The technique is interesting and there is a strong case for the proposed method being more practical than quantum illumination schemes, but there are a number of points I believe should be clarified before the work is considered for publication.

- Over what distances is the ranging performed? Is this still comparable to experiments with entangled photons?
- Can the authors also comment on the effective ranging distance that is practically achievable? As an optical delay line is required I would expect this to be quite limiting. Is it assumed that the distance is known a priori?
- The authors mention that dispersion compensation is required, is this due to specifically dispersion in the fibers? Does dispersion from propagation in air need to be taken into account (and again does the distance need to be known a priori if so)?
- In the introduction: "the performance advantage of QI over classical detection remarkably survives through an entanglement-breaking environment of high noise and loss". I find this a little misleading, these schemes still work precisely because the energy-time entanglement is still present, the environment is therefore not "entanglement-breaking"
- In the abstract: "of completely indistinguishable in-band noise", I think this needs to be more clearly defined. If the signal can be filtered out then by definition it is not "completely indistinguishable", but still can be indistinguishable in certain degrees of freedom.
- I find the name Qins LiDAR also potentially a bit misleading as there are already other approaches to LiDAR that are quantum inspired but different in concept e.g. using Hong-Ou-Mandel interferometry
- What is the bandwidth of the pump laser? Would this not determine the linewidth of the c-SFG signal and therefore the efficiency of the spectral filtering?
- LEDFA and CDEFA appear undefined in fig 2
- It's not 100% clear to me why the homodyne detection is required. The key principle

proposed seems to be the c-SFG filtering and the homodyne detection is acting as an additional filter on top of this. In which case, how much noise rejection is actually being performed by the c-SFG filter and how much by the homodyne detection?

- Does the c-SFG filter effect the ranging resolution or is this solely determined by the probe bandwidth?

Reviewer #2 (Remarks to the Author):

The authors present a technique for the optical measurement of distances using difference and sum frequency generation of a narrowband laser and an ASE source. The technique demonstrates 100 dB in band noise rejection and high distance resolution. The key for the high noise rejection is that the noise and reference light after difference frequency generation of a narrowband laser with the ASE source have conjugated phase and are recombined into a copy of the CW pump light at the correct time delay. The authors demonstrate the rejection of in band optical noise of 100 dB, which is achieved by narrowband optical and electrical filtering. Here a 10 Hz electrical filter is used in conjunction with 1 THz of optical bandwidth to achieve 100 dB noise suppression.

In summary, I cannot endorse the paper for publication in Nature Communications in its current form.

The technique of using sum frequency generation, here referred to as “quantum-inspired”, of time-frequency anticorrelated light to improve the visibility in an optical interferometer was first introduced by R. Kaltenbaek et al., Nature Physics 4, 864 (2009) using anti-chirped optical pulses.

The presented technique of quantum inspired LiDAR is quite closely related to random modulation LiDAR with coherent detection. Hence, it should be possible to not only detect the distance but also the velocity of the object using spectral cross correlation. I would strengthen the applicability of the proposed technique in comparison to existing LiDAR schemes such as FMCW.

I also find the claim about the near term use of the presented technique for automotive

LiDAR exaggerated for the following reasons:

- 1) There is no indication that the mutual interference of properly gated TOF and FMCW sensors is a prohibitive feature for automotive LiDAR. The proposed technique is much more technically challenging.
- 2) The proposed technique requires a form of mechanical delay scanning which seems quite impossible to implement for a 0-200m delay range on a cm-sized LiDAR sensor.
- 3) The acquisition speed is many orders of magnitude too low. Here it takes 100s of ms to scan 4mm of length on a single pixel. A realistic LiDAR sensor (300.000 kPix/frame @ 15 fps) requires 4.5 MPix/s. How would this be implemented in the present technique given that the noise rejection depends on the ratio of optical and electrical bandwidth?
- 4) The current implementation of the technique already requires 250 mW of optical power to detect a 4mm long segment of a single pixel. How can the total system power be scaled below 1 W for a realistic detection scenario?

In my opinion, the presented measurement technique is much more akin to optical coherence tomography (OCT) than to automotive LiDAR, because of its limited distance range and high distance resolution, which is mandated by the large optical bandwidth to achieve noise rejection. However, OCT usually does not suffer from noise light injection like automotive LiDAR. Is there a way the distance can be measured without mechanical delay scanning?

In the second part of the paper, a numerical argument is given for the usefulness of chaotic time frequency modes for quantum information processing. However, this argument remains underdeveloped and no experimental investigation of the application of chaotic time frequency modes for quantum information processing is performed and no comparison with the state of the art is given.

The quantum inspired LiDAR uses a single a single chaotic mode defined by the ASE source and the seeded DFG process. In a quantum light source, unseeded DFG is used for photon pair generation or squeezing. It remains unclear from the manuscript, how a seeded DFG can be used as a quantum light source.

I find that the two sections of the paper are poorly linked given that the Lidar experiment

and demonstration is a purely classical experiment and has no direct relation with the encoding of quantum information in single photon pulses.

Please find a few comments on the presentation of the manuscript figures:

Figure 4a: Please indicate the broken x-Axis properly.

Figure 4b: Is the spectrum normalized? If so, please indicate in the caption.

Reviewer #3 (Remarks to the Author):

In this work, the authors present an innovative approach to enhance the noise rejection capability of LIDAR using non-linear optical methods and a coherent detection. In the presence of a pump and reference signals, a probe is generated through difference-frequency generation. The probe illuminates a target is recombined with the reference in a sum-frequency generation (SFG) configuration, the output of which is filtered using a narrow bandpass filter, and measured using a homodyne detector. The narrow bandpass filter is centred around the emission peak of coherent SFG and reduces the background incoherent SHG that is simultaneously generated. By measuring the ratio of a back-scattered probe to that of a noise illumination, the authors have shown a noise rejection of over 100 dB.

The technique is rather elegant, the results are well presented and their discussion is clear and well elaborated. The manuscript is also well written and largely free of typos. I believe the present work will be of interest to the optical sensing community at large and should be considered for publication in Nature Communications. I will however invite the authors to first address the following questions and comments:

1) what is the spectral bandwidth of the DFG and SFG processes in the PPLN waveguides?

2) what is the spatial resolution of the telescope at the plane of the targets?

3) The authors attribute the multi-peak structure in Fig. 5a to different paths inside the receiving telescope. Could the authors justify based on the telescope structure what these

paths are? How would these peaks differ from back-reflection from structures at various depths within the field of view of the telescope?

4) The depth resolution is inversely proportional to the bandwidth of the probe, but I would think that the effective bandwidth will be limited by the phase matching through the relatively long 5cm SFG crystal. Is that not the case here?

5) In Equation 10, why is coherent SFG efficiency an oscillating function of the probe photon flux?

6) Could the authors provide some more details regarding the waveguides [propagation losses, coupling efficiency, AR coating if any on the faces, phase matching type (0, 1 or 2)]?

7) There is a note on top of Fig. 4a that reads "need to work on the figure later"

Reply to reviewer 1

We are delighted to hear that the reviewer acknowledge the novelty of this work and we thank the reviewer for providing useful revision suggestion. We have now revised the manuscript based on the reviewer's suggestion. In this document, we provide point-by-point responses to the reviewer's concerns, in the hope that the reviewer could reconsider the publication of this manuscript in *Nature Communication*.

- **Comment:** Over what distances is the ranging performed? Is this still comparable to experiments with entangled photons?
- **Reply:** We thank the reviewer for reminding us of highlighting the ranging information. The target distance in our current implementation is around 4 meters (8 meters round trip) away from the telescope transceiver. This is only limited by our lab room space since a significant target detection signal (10dB above the shot noise floor) from a weakly reflecting target can still be observed at this distance. It is also worth mentioning that our system transmission efficiency (insertion loss of power monitoring elements, waveguide coupling loss, optical filter loss, telescope inefficiency) is currently far from optimum due to constraints of our proof-of-principle implementation. Once these technical limitations are improved with systematic re-factoring and optimization, our protocol is expected to achieve the same sensitivity and detection range as other classical LiDARs with single photon detection.

To answer the reviewer's question directly, there has been no demonstration of target detection protocols with entangled light sources that provide a larger free space detection range or higher sensitivity, to the best of our knowledge. Although there have been target detection protocols using classical sources and single photon detection to achieve a long detection range, their noise resilience is not comparable to the unprecedented $> 100\text{dB}$ rejection of in-band noise demonstrated here.

- **Comment:** Can the authors also comment on the effective ranging distance that is practically achievable? As an optical delay line is required I would expect this to be quite limiting. Is it assumed that the distance is known a priori?
- **Reply:** The reviewer raised a valid concern about the scanning range of our protocol: the free space mechanical delay used in the current setup can be a performance limiting factor for detection range and it requires the target location to be known approximately a priori. However, this is not a fundamental limit to our protocol as solid-state optical switches can also provide delay scanning capabilities with significant enhancement in terms of scanning speed and range.

To directly address the concern of the reviewer, we purposely built a new setup of solid-state optical delay to showcase the possibility of fast, long-range scanning, without using mechanical delay lines. It is implemented with two magneto-optical switches (1×4 channel) and two mems (1×8 channels) optical switches, as shown in Fig.1. The total scanning range is $4\times 4\times 8\times 1\text{mm} = 12.8\text{cm}$, which is only limited by the number of optical switches and the number of channels of each switch. The switching speed of our switch is around $10\mu\text{s}$ for the magneto-optical switch and 0.5ms for the mems switch. The insertion losses of switches are compensated by a fiber amplifier.

The ranging result with this new delay line is shown in Fig. 1(b). As can be seen, the location of the target can be determined with mm-level resolution. In light of this result, we project that with more switches of the same type, the delay can be scanned through 250 meters (with 1mm step) within 2.5 seconds. However, this is not yet the state of art speed for commercially available optical switching

products. We anticipate that the scan time can be further decreased by a factor of 200 should we use a different model of solid-state optical switch ¹.

Figure 1: (a) the schematic of the non-mechanical delay line: circ1,circ2: optical circulators, r1-r8: optical retro-reflector, s1,s2: solid state optical switches, m1,m2: mems optical switches (only four channels are shown for each switch are shown for simplicity, amp: low power erbium-doped fiber amplifier module. The relative (fiber optical) delay of different channels for s1 and s2 are 1mm and 4mm, respectively. The relative (fiber optical) delay of different channels for the mems switch pair is 1.6cm. (b) the experimental ranging result. Different path indices indicate different combinations of optical switch channels. Adjacent channels have around 1mm (fiber optical) path length difference. The target detection signal is measured in a log scale using balanced homodyne detection and spectrum analyzer.

- **Comment:** The authors mention that dispersion compensation is required, is this due to specifically dispersion in the fibers? Does dispersion from propagation in the air need to be taken into account (and again does the distance need to be known a priori if so)?
- **Reply:** We agree with the reviewer that group velocity dispersion from the optical fiber is indeed an important concern that has a significant impact on the system sensitivity. We confirm that the dispersion compensation is to compensate for the fiber dispersion within the system (mainly the fiber amplifiers). If fiber dispersion is not properly compensated, the sum frequency generation efficiency will be severely degraded due to destructive nonlinear interference. The total amount of dispersion compensated is 1.6ps/nm, which corresponds to around 89 meters of fiber. On the other hand, the dispersion of air (23.5×10^{-6} ps/nm/km at 1550nm) is 6 orders of magnitude lower than the dispersion of optical fiber (18ps/nm/km). Thus we expect it to not be a performance limiting factor within the kilometer level of ranging distance.
- **Comment:** In the introduction: "the performance advantage of QI over classical detection remarkably survives through an entanglement-breaking environment of high noise and loss". I find this a little misleading, these schemes still work precisely because the energy-time entanglement is still present, the environment is therefore not "entanglement-breaking"

¹NanoSpeed™ Series, Agiltron

- Reply: For quantum illumination with Gaussian states, the initially entangled probe and reference light loses entanglement after the noisy and lossy target detection channel. However, it has been shown that such a received state can still provide detection accuracy enhancement that is not attainable with classical light only[1]. However, it remains not clear yet whether this argument applies to other quantum illumination protocols such as ones based on time-frequency entanglement[2]. To avoid confusion, we decide to remove this ambiguous statement from the text.
- Comment: In the abstract: "of completely indistinguishable in-band noise", I think this needs to be more clearly defined. If the signal can be filtered out then by definition it is not "completely indistinguishable", but still can be indistinguishable in certain degrees of freedom.
- Reply: The reviewer is correct on this point. The in-band noise is indeed distinguishable and filterable if one has its time-frequency correlation information, which is the reference light that is only accessible to the transceiver station. To clarify this, we have changed the phrase to "of indistinguishable (have identical statistical properties in every degrees of freedom) in-band noise". We thank the reviewer for making this suggestion.
- Comment: I find the name Qins LiDAR also potentially a bit misleading as there are already other approaches to LiDAR that are quantum inspired but different in concept e.g. using Hong-Ou-Mandel interferometry
- Reply: We agree with the reviewer that we should avoid potential confusion with other quantum-inspired target detection techniques. Therefore, we have changed the acronym to "chaotic Quantum Frequency Conversion" Lidar.
- Comment: What is the bandwidth of the pump laser? Would this not determine the linewidth of the c-SFG signal and therefore the efficiency of the spectral filtering?
- Reply: The pump laser used is a Coherent MBR110 Ti-Sapphire laser that features 75kHz linewidth. This linewidth is not currently affecting the phase-sensitive measurement of SFG light because the generated c-SFG and local oscillator light have identical phase fluctuation that cancels each other in the balanced homodyne detection. However, it should be noted that when the target distance is beyond the coherence length (4km) of the pump laser, the c-SFG light and local oscillator light will become relatively incoherent and the target detection signal will be spectrally broadened, resulting in degradation of detection sensitivity. We have made this point more clear in the manuscript now.
- Comment: It's not 100% clear to me why the homodyne detection is required. The key principle proposed seems to be the c-SFG filtering and the homodyne detection is acting as an additional filter on top of this. In which case, how much noise rejection is actually being performed by the c-SFG filter and how much by the homodyne detection?
- Reply: The reviewer is correct that in the current setup, homodyne detection is not strictly needed. The principle of noise rejection stems from the large bandwidth ratio between the broadband probe and noise light and the single frequency c-SFG light. Electrical domain filtering only contributes additional filtering after most of the noise is optically filtered. The difference between the two filtering mechanism is as follow. Optical filtering is responsible for reducing the noise power (i-SFG) before photon detection to avoid detector nonlinearity and saturation. Electrical filtering can then provide additional filtering to effectively resolve the target detection signal in the data processing stage. In the current setup, the optical filtering of i-SFG is limited by the implementation: the SFG bandwidth after the waveguide and the diffraction grating is 0.12nm FWHM. This corresponds to around 97% filtering of i-SFG power that would otherwise be generated within around 3.75nm bandwidth (FWHM, corresponding to 7.5nm probe and reference bandwidth around 1560nm). Should we use a commercially available Fabre-Perot filter ² that features around 1MHz bandwidth, the total optical filtering of i-SFG will be 63dB.
- Comment: LEDFA and CDEFA appear undefined in fig 2

²SA30 series Fabre Perot filters, Thorlabs. Inc.

- Reply: We thank the reviewer for pointing this out and we have now properly defined these acronyms in the manuscript. To confirm, LEDFA and CEDFA stand for L and C band erbium doped fiber amplifier, respectively.
- Comment: Does the c-SFG filter effect the ranging resolution or is this solely determined by the probe bandwidth?
- Reply: The reviewer is correct that the ranging resolution is solely limited by the probe bandwidth. Since the target distance information is only encoded in the power of c-SFG light, filtering of i-SFG will have no effect on the ranging resolution.

References

- [1] Si-Hui Tan, Baris I Erkmen, Vittorio Giovannetti, Saikat Guha, Seth Lloyd, Lorenzo Maccone, Stefano Pirandola, and Jeffrey H Shapiro. Quantum illumination with gaussian states. *Physical review letters*, 101(25):253601, 2008.
- [2] Han Liu and Amr S Helmy. Joint measurement of time–frequency entanglement via sum frequency generation. *npj Quantum Information*, 6(1):1–6, 2020.

Reply to reviewer 2

We thank the reviewer for taking the time to review and evaluate our manuscript and provide valuable feedback. We have now made substantial edits to the manuscript taking into account the feedback which enables us to more consistently present/explain the ideas and results as well as provide relevant context of our work. In this document, we list point-by-point responses to the reviewer’s concerns, in the hope that these improvements would render the manuscript acceptable for publication from the reviewer’s standpoint.

- **Comment:** The technique of using sum frequency generation, here referred to as “quantum-inspired”, of time-frequency anti-correlated light to improve the visibility in an optical interferometer was first introduced by R. Kaltenbaek et al., *Nature Physics* 4, 864 (2009) using anti-chirped optical pulses.
- **Reply:** The reviewer is correct that the concept of chaotic mode conversion and chirped pulse interferometry share the common denominator of utilizing sum-frequency generation to analyze time-frequency anti-correlation. Nevertheless, we would like to stress that it is the distinctions of this protocol from other existing correlation-based protocols that enable the unprecedented high sensitivity and noise resilience we demonstrate here.

First, the type of time-frequency correlation used in this setup is fundamentally different from that used in chirped pulse interferometry. In chirped pulse interferometry, time-frequency correlations are created by manually applying an equal amount of opposite dispersion on ultrafast laser pulses. Our technique, on the other hand, generates a time-frequency correlation by taking advantage of the DFG energy conservation law and the fundamentally chaotic nature of ASE. This is closely parallel to the generation of strong time-frequency entanglement in SPDC but at a much higher classical power level. As such, our time-frequency correlation (picosecond level correlation time and hertz level correlation bandwidth) is not only much stronger than what is tenable by manually applying dispersions to ultrafast pulses but also comes in a much simpler and robust setup only using fiber communication components and a nonlinear waveguide.

Second, this work highlights the high quantum efficiency of SFG for coherent measurement of time-frequency correlation: with sufficiently narrowband SFG phase matching, correlated probe light can be 100% converted. While high efficiency may not be as crucial for OCT applications that do not have high noise levels, it is crucial for target detection applications to achieve a long detection range or high sensitivity. The high quantum efficiency is also an essential requirement of quantum information processing, which will be discussed later in this response.

In the revised document we have now highlighted the similarities and differences between this work and chirped ultrafast laser pulse interferometry.

- **Comment:** The presented technique of quantum-inspired LiDAR is quite closely related to random modulation LiDAR with coherent detection. Hence, it should be possible to not only detect the distance but also the velocity of the object using spectral cross-correlation.
- **Reply:** Indeed, the use of the classically correlated probe and reference light is closely related to classical random modulation LiDAR that either takes random electrical waveform as an input[1] or uses the chaotic dynamics of a semiconductor laser[2]. But there are two essential differences that have nontrivial and substantial implications on the performance advantage of our system: (1) the use of SFG instead of linear interference-based coherent detection (please see the next response for a detailed discussion) and (2) the use of broadband optical random waveform (ASE fluctuation) that features over

Figure 1: The experimentally measured Doppler shift of the c-SFG frequency when the target delay is scanning. Resolution bandwidth: 1Khz.

1000GHz bandwidth. Such bandwidth is not achievable with classical random modulation LiDAR.

The reviewer is also correct that our target detection system is capable of measuring not only distance but also the target velocity. In fact, we confirmed the existence of a 764 Hz Doppler shift of the measured target detection signal when the target delay is being scanned with around 1.2m/s velocity speed. New measurements have been carried out and included in the revised manuscript to demonstrate this.

In the revised document we have now highlighted the similarities and differences between this work and random modulation LiDAR. We also added a section describing the velocity measurement.

- **Comment:** There is no indication that the mutual interference of properly gated TOF and FMCW sensors is a prohibitive feature for automotive LiDAR. The proposed technique is much more technically challenging.
- **Reply:** It is true that jamming and mutual interference may not be a prohibiting factor for civil applications like automotive LiDAR in a properly regulated environment. Nevertheless, we would like to emphasize that this paper is not about reporting just another LiDAR modality. One of the motivations of the work is the substantially enhanced capabilities for target detection such as unprecedented noise resilience that is not reported even using a nonclassical state of light (which is ten orders of magnitude weaker in power). The capabilities offered here including noise resilience can enable many target detection protocols but are not at all in any way restricted to LiDAR. It is neither constructive to take the results from this initial report and set them in a one-to-one comparison with existing commercial systems which offer similar functions, such as conventional LiDARs. There are numerous modifications and enhancements that can be added to this work reported here beyond the first report and demo, to either extend its functionality and systematic optimization. For example, the current system can be reimplemented with only fiber optical components in a much more compact and robust factor.

That being said, although conventional LiDARs based on ToF and FMCW may tolerate a certain level of noise power, especially under controlled environments and in civil applications, such noise resilience is fundamentally limited for the following reason. The conventional receiver structure allows direct interaction between high-power noise light and the photodiode, which leads to detection nonlinearity at best and completely blinding/damaging of the LiDAR at worst, both leading to severe safety concerns. Although conventional gating and filtering techniques can reduce the impact of out-of-band noise background to a certain degree, they are completely ineffective against active jamming sources that are intentionally prepared to have identical characteristics as the authentic probe source.

The technique reported here completely avoids the aforementioned limitation of conventional LiDAR: (1) by using an SFG receiver, noise light is filtered before being seen by the photodiode (2) the chaotic nature of probe light makes it impossible to create “indistinguishable jamming source”. An additional advantage that may be of use in certain scenarios is that the broadband chaotic probe light is covert (cannot be distinguished from background noise light) and cannot be detected by other parties.

- **Comment:** The proposed technique requires a form of mechanical delay scanning which seems quite impossible to implement for a 0-200m delay range on a cm-sized LiDAR sensor.
- **Reply:** It is helpful if the reviewer recognizes that this is the first report of a novel technique that offers unparalleled performance metrics. While it has some similarities to exiting approaches such as OCT and Chaotic LiDAR, it also offers distinct difference and advantages which are witnessed in the performance measured. The paper is not a report on the latest flavour of LiDAR implementation. In this initial demonstration, we used a tunable mechanical delay line to demonstrate the ranging capability. This is only limited by the resources that are available to our academic group at the moment. However, we would like to take this opportunity to emphasize that the tunable delay line does not have to be mechanical at all and can achieve fast scanning over a long range.

To demonstrate the long-range fast scanning capability of our target detection receiver, we have recently just implemented a solid state scanning delay line using a borrowed kit of two magneto-optical optical switches (1×4 channel) and two mems (1×8 channels) optical switches, as shown in Fig.2. The total scanning range is $1\text{mm} \times 4 \times 4 \times 8 = 12.8\text{cm}$, which is only limited by the number of optical switches and the number of channels of each switch. The switching speed of our switch is around 10us for the solid state switch and 0.5ms for the mems switch. The insertion losses of switches are compensated by a fiber amplifier.

The ranging result with this new delay line is shown in Fig. 2(b). As can be seen, the location of the target can be determined with mm-level resolution. In light of this result, we project that with more switches of the same type, the delay can be scanned through 250 meters (with 1mm step) within around 2.5 seconds. However, this is not yet the state of art speed for commercially available optical switching products. We anticipate that the scan time can be further decreased by a factor of 200 should we use a different model of solid-state optical switch ¹.

- **Comment:** The acquisition speed is many orders of magnitude too low. Here it takes 100s of ms to scan 4mm of length on a single pixel. A realistic LiDAR sensor (300.000 kPix/frame @ 15 fps) requires 4.5 MPix/s. How would this be implemented in the present technique given that the noise rejection depends on the ratio of optical and electrical bandwidth?
- **Reply:** In this work, we are reporting a novel approach to an important class of measurement with desirable performance that is not yet available from commercial systems. The current implementation, however, is limited by the resource available to our academic group and is focused on the noise resilience and sensitivity aspects of target detection (which constitute the main result of this report). As such, we believe it is not constructive to directly compare this initial proof of principle demonstration (with little engineering optimization) to highly engineered commercial systems in all different aspects. Nevertheless, we also believe that the reviewer’s concern with regard to the trade-off between imaging resolution and noise resilience can be addressed with adequate augmentation to the current system implementation:

(1) in this initial report, we focus on analyzing the generation and measurement of strong time-frequency correlation and its potential application in target detection. Therefore, we only use a single spatial mode target detection transceiver that resembles a single-pixel camera. Imaging operation can still be done through raster scanning with a galvo mirror at the cost of reducing the integration time of each pixel. Even so, the noise resilience for 1us integration time per pixel can reach up to over 50dB, which

¹NanoSpeed™ Series, Agiltron

Figure 2: (a) the schematic of the solid-state delay line: circ1,circ2: optical circulators, r1-r8: optical retroreflector, s1,s2: solid state optical switches, m1,m2: mems optical switches (only four channels are shown for each switch for simplicity, amp: low power erbium-doped fiber amplifier module). The relative (fiber optical) delay of different channels for s1 and s2 are 1mm and 4mm, respectively. The relative (fiber optical) delay of different channels for the mems switch pair is 1.6cm. (b) the experimental ranging result. Different path indices indicate different combinations of optical switch channels. Adjacent channels have around 1mm (fiber optical) path length difference. The target detection signal is measured on a log scale using balanced homodyne detection and spectrum analyzer.

is still beyond what is achievable with commercial LiDAR modalities.

(2)The target detection protocol reported here is not necessarily constrained to single spatial mode operation. It is possible to deploy a multi-spatial mode receiver that collects probe light in different spatial modes and then performs independent SFG analysis at the same time. In such a setup, the spatial resolution of detection will be only dictated by the number of independent spatial modes without the need for raster scanning. This approach can be realized through image upconversion inside a nonlinear crystal or array of nonlinear waveguides.

- **Comment:** The current implementation of the technique already requires 250 mW of optical power to detect a 4mm long segment of a single pixel. How can the total system power be scaled below 1 W for a realistic detection scenario?
- **Reply:** The reviewer raised an important concern about the system’s efficiency and practicalness. To address this concern, we would like to first clarify that the 250mW is the reference light power that stays internal to the target detection system. The actual probe power emitted from the transceiver to interact with the target object is around 50mW, of which only around -110dBm of collected probe power is converted through the SFG process. In fact, the ability to resolve such a weak signal in a high noise background critically depends on the high quantum efficiency of the SFG receiver, which is measured to be more than 90% as shown in Fig. (3b) in the main text. Nevertheless, it is true that various sources of linear loss (Table. 1) within the system can be further improved to achieve higher detection sensitivity. Once these losses are improved with systematic refactoring and upgraded optics, the overall system efficiency will be comparable to commercial LiDARs that feature single photon sensitivity.

source of loss	waveguide coupling	fiber interconnection	power monitoring	telescope inefficiency	freespace optics
magnitude	3dB	1dB	4dB	10dB	1.35dB

Table 1: Summary of different sources of loss of the experimental setup. Power monitoring refers to the insertion of a 50:50 power splitting of the SFG light to enable simultaneous intensity and balanced homodyne measurement. The telescope efficiency is measured with a perfectly reflecting mirror placed at 4meters away.

- **Comment:** In my opinion, the presented measurement technique is much more akin to optical coherence tomography (OCT) than to automotive LiDAR, because of its limited distance range and high distance resolution, which is mandated by the large optical bandwidth to achieve noise rejection. However, OCT usually does not suffer from noise light injection like automotive LiDAR. Is there a way the distance can be measured without mechanical delay scanning?
- **Reply:** This proof of principle demonstration is aimed at providing a novel solution to an important problem (suppress overwhelming noise/jamming power while still retaining near unity quantum efficiency), that has important application in many areas, including but not limited to, automotive LiDAR. The particular applications, where this work proves most impactful will certainly depend on the work that will be carried out beyond this initial report. We also agree with the reviewer that using broadband optical correlation is similar to OCT. However, as the reviewer has noted earlier, sensing applications like LiDARs suffer from a much higher level of noise light than OCT. The problem is addressed by our SFG receiver which can efficiently suppress background/jamming noise without impacting the ranging accuracy and sensitivity. Such noise/jamming suppression cannot be achieved by conventional interferometric detection that does not feature optical domain noise rejection.

The possibility of using solid-state delay for fast long-range scanning can be found in previous discussions.

- **Comment:** In the second part of the paper, a numerical argument is given for the usefulness of chaotic time-frequency modes for quantum information processing. However, this argument remains underdeveloped and no experimental investigation of the application of chaotic time frequency modes for

quantum information processing is performed and no comparison with the state of the art is given. The quantum inspired LiDAR uses a single chaotic mode defined by the ASE source and the seeded DFG process. In a quantum light source, unseeded DFG is used for photon pair generation or squeezing. It remains unclear from the manuscript, how a seeded DFG can be used as a quantum light source.

I find that the two sections of the paper are poorly linked given that the Lidar experiment and demonstration is a purely classical experiment and has no direct relation with the encoding of quantum information in single photon pulses.

- Reply: We apologize for the confusion that may have been caused in the discussion section. The second part of the manuscript serves to discuss the possibility of how this technique, when developed in a different direction, can be of immense use in quantum optics. This is possible because of the following conceptual similarity: to coherently separate probe light from in-band noise light is similar to the coherently separate quantum state of light from a particular time-frequency mode from other orthogonal modes. The latter protocol is known as quantum frequency conversion (QFC)[3].

To clarify the reviewer’s concern, the classical probe and reference light generated in the seeded DFG process is **not** quantum light. Instead, they define a conjugate pair of probes and reference time-frequency modes with their waveform. Quantum light (generated through unseeded DFG) in such probe and reference modes can be measured in one of the following ways:

- **Homodyne detection:** probe (reference) light can be used as the local oscillator light for homodyne detection of quantum light in the probe (reference) mode.
- **QFC:** probe (reference) light can be used as the QFC pump light to selectively convert quantum light in the reference (probe) mode to another frequency. This corresponds to using the strong reference light to convert the weak probe light in our SFG receiver.

Using a chaotic probe and reference to define a time-frequency mode has several-fold advantages for QFC. Interestingly, this is conceptually related to the target detection performance advantage we demonstrate:

- The high efficiency of chaotic-mode-QFC is equivalent to the high SFG conversion efficiency of our target detection receiver
- The low crosstalk of chaotic-mode-QFC from irrelevant modes is equivalent to the high in-band noise rejection of our target detection receiver.
- Unseeded DFG process naturally generated two mode entangled light in the chaotic probe and reference mode. In other words, chaotic probe and reference modes generated in seed DFG coincide with the intrinsic quantum modes of unseeded DFG light (squeezed vacuum).

We thank the reviewer for providing feedback on this section. We have now improved the discussion accordingly to avoid confusion. We also make a more direct comparison between chaotic mode based QFC and established pulse light based QFC [3] to provide more sufficient context.

- Comment: Figure 4a: Please indicate the broken x-Axis properly. Figure 4b: Is the spectrum normalized? If so, please indicate in the caption.
- Reply: We thank the reviewer for pointing out these. The manuscript has been modified accordingly.

References

- [1] Liyan Feng, Huazheng Gao, Jianxun Zhang, Minghai Yu, Xianfeng Chen, Weisheng Hu, and Lilin Yi. Fpga-based digital chaotic anti-interference lidar system. *Optics Express*, 29(2):719–728, 2021.

- [2] Fan-Yi Lin and Jia-Ming Liu. Chaotic lidar. *IEEE journal of selected topics in quantum electronics*, 10(5):991–997, 2004.
- [3] Benjamin Brecht, Dileep V Reddy, Christine Silberhorn, and Michael G Raymer. Photon temporal modes: a complete framework for quantum information science. *Physical Review X*, 5(4):041017, 2015.

Reply to reviewer 3

We are delighted to hear that the reviewer acknowledge the novelty of this work and we thank the reviewer for providing useful revision suggestion. We have now revised the manuscript based on the reviewer's suggestion. In this document, we provide point-by-point responses to the reviewer's concerns, in the hope that the reviewer could reconsider the publication of this manuscript in *Nature Communication*.

- **Comment:** What is the spectral bandwidth of the DFG and SFG processes in the PPLN waveguides?

Figure 1: The spectrum of DFG and SHG for characterization of $\Delta\lambda_{SH}$ and $\Delta\lambda_{FH}$. For the DFG spectrum, the signal wavelength is varied, and the conversion efficiency of the idler light is measured. As can be seen, efficient DFG can happen over $\Delta\lambda_{FH}$ 34nm range of signal and idler bandwidth. For the SHG spectrum, the relative conversion efficiency is plotted as the function of SHG wavelength. As can be seen, efficiency SHG can happen over a range of $\Delta\lambda_{SH} = 0.24\text{nm}$ around 781.8nm

- **Reply:** The $\chi^{(2)}$ nonlinear process (both DFG and SFG) have two characteristic bandwidths :

phasematching bandwidth for the second harmonics $\Delta\lambda_{SH}$: The wavelength range in which SFG light can be efficiently generated. This is the same as the wavelength range of DFG pump light that allows for efficient DFG conversion. This value is experimentally measured to be 0.24nm (full-width half max) around 781.8nm, as can be seen in Fig.1(a).

phasematching bandwidth for the fundamental harmonics $\Delta\lambda_{FH}$: The wavelength range of probe light that allows for efficiency SFG conversion (provided that the generated SFG light is within $\Delta\lambda_{SH}$). This is the same as the wavelength range of DFG signal light that allows for efficiency DFG conversion (provided that the DFG pump light is within $\Delta\lambda_{SH}$). This value is experimentally measured to be 34 nm (3dB), as can be seen from Fig.1(b).

We would like to take this opportunity to highlight the impact of $\Delta\lambda_{SH}$ and $\Delta\lambda_{FH}$ on the target detection performance metrics. The bandwidth $\Delta\lambda_{SH}$ needs to be as narrow as possible to only allow coherent SFG while rejecting most of the incoherent SFG. The bandwidth $\Delta\lambda_{FH}$ needs to be as broad as possible to allow larger probe (reference) light bandwidth, which is inversely proportional to the distance resolution.

- **Comment:** what is the spatial resolution of the telescope at the plane of the targets?
- **Reply:** The transverse resolution of our free space transceiver is dictated by the probe light spot size on the target object, which is measured to be around 20mm in diameter (Fig. 2(a)). The resolution

Figure 2: The beam size of the probe light on the target object when the probe light is coming from the (a) collimator and (b) the telescope (injection light into the output port). The beam size is measured using a moving razor blade that partially blocks the beam.

of the telescope is measured by injecting light into the telescope output and measuring the beam spot diameter (around 30mm, Fig.2(b)) on the target object. We expect that with an upgraded telescope and collimator structure, the transverse resolution can be adjusted to suit the needs of different applications.

- **Comment:** The authors attribute the multi-peak structure in Fig. 5a to different paths inside the receiving telescope. Could the authors justify based on the telescope structure what these paths are? How would these peaks differ from back-reflection from structures at various depths within the field of view of the telescope?
- **Reply:** The reviewer raised a valid concern about the origin of the multiple side peaks of the ranging scan. After further investigation of the experimental setup, we found evidence that renders the multi-reflection hypothesis less likely, as compared to other possible causes. This is because the multi-reflection path inside the telescope corresponds to a much larger delay (the minimum separation between different lenses is 15mm, as shown in Fig. 3) than the separation of observed ranging peaks. Another possibility that can potentially lead to multiple ranging peaks is the different path that reflected probe light take before and within the telescope. As the reviewer suggests, the surface of the target object may be not equidistant to the receiving telescope. This may lead to different probe path length difference corresponding to different depths of view. Besides this, the diffusively reflecting target object allows for different directions of reflection, resulting in a large beam diameter > 2inch at the first surface of the telescope. Then probe light with different distances from the optical axis may be focused differently due to the spherical aberration. To confirm that the multiple ranging peaks indeed correspond to different probe paths, the transceiver is tested with a silver mirror as the object that only allows specular reflection. The ranging result then only shows a single peak corresponding to a single propagation path. Another possibility that can potentially lead to multiple ranging peaks

Figure 3: The schematic of the telescope structure. The lenses used (from left to right) are Thorlabs LC1093-C-ML, LC1315-C-ML, LA1401-C, LC1054-C, LA1540-C. The surface-to-surface distances between adjacent lenses are 4.7mm, 35mm, 40mm, 15mm, respectively.

is the multiple paths that probe light takes between the target object and the telescope and within

the telescope. As the reviewer suggests, the surface of the target object may be not equidistant to the receiving telescope. This may cause different path length difference corresponding to different depths of view. Besides this, the diffusively reflecting target object allows for different directions of reflection, resulting in a large beam diameter > 2 inch at the first surface of the telescope. Then probe light with different distances from the optical axis may be focused to different depths due to spherical aberration. As a comparison test, we tried using a silver mirror as the object that only allow specular reflection. The ranging result then only shows a single peak corresponding to a single propagation path.

We thank the reviewer for pointing this out and we have now revised the document accordingly with the modified conclusion.

- **Comment:** 4) The depth resolution is inversely proportional to the bandwidth of the probe, but I would think that the effective bandwidth will be limited by the phase matching through the relatively long 5cm SFG crystal. Is that not the case here?
- **Reply:** The reviewer is correct that the distance resolution may be affected by SFG phase matching. To be more specific, the distance resolution is limited by either the probe light bandwidth or phase matching bandwidth $\Delta\lambda_{FH}$, whichever is lower. It is true that $\Delta\lambda_{FH}$ decreases with a longer waveguide, a 5cm long waveguide still allows for more than 34nm of $\Delta\lambda_{FH}$ as can be seen in Fig. 1. This is because around 1550nm, the group velocity walk-off in lithium niobate between probe and reference light (assume 20nm separation) is relatively small (88fs for 5cm long waveguide). As a result, instead of the phase matching bandwidth $\Delta\lambda_{FH}$, the probe light bandwidth is the actual limiting factor of distance resolution in the current setup.

We thank the reviewer for raising the concern. We have now revised the document to state this fact more explicitly.

- **Comment:** In Equation 10, why is coherent SFG efficiency an oscillating function of the probe photon flux?
- **Reply:** The reviewer raised an interesting question about the dependency of the SFG efficiency on the reference light power. This reason for SFG to start to decrease in efficiency is that SFG is a photon number preserving process: SFG can only convert probe photon to SFG and vice versa with the total probe and SFG photon number preserved. To illustrate this, consider theoretically a long SFG waveguide with strong reference light and weak probe light at the input (Fig. 4). After a certain interaction length, the probe light is completely converted to SFG light. Then at that point, the nonlinear interaction can only be a DFG process that depletes the SFG light to generate probe light, since no probe photon is available to participate in SFG.

Figure 4: The illustration of SFG-DFG transition as a function of nonlinear interaction length. “Maximal SFG” marks the interaction length that corresponds to the complete SFG conversion of the probe photons, after which DFG interaction will create probe photons by consuming SFG photons.

We have now included comments about this behavior in the revised document.

- Comment: Could the authors provide some more details regarding the waveguides [propagation losses, coupling efficiency, AR coating if any on the faces, phase matching type (0, 1 or 2)]?
- Reply: We have now added a section in the supplementary material to provide detailed specifications of the waveguide, including propagation and coupling loss, AR coating bandwidth, and phase matching type. We thank the reviewer for the suggestion.
- Comment: There is a note on top of Fig. 4a that reads "need to work on the figure later"
- Reply: We thank the reviewer for pointing out this mistake. We have now replaced the figure with an updated version.

REVIEWERS' COMMENTS

Reviewer #1 (Remarks to the Author):

The authors have addressed my concerns about the work satisfactorily, however I believe that several of the points should be more directly addressed in the manuscript.

- The range used should be included
- A discussion of the optical delay line should be included (I think the author's response to this point would be suitable for the Supplementary Material)
- The dispersion should be mentioned. Although I agree with the authors' reply that dispersion in air over kilometre distances will minimally impact the results, kilometres of fiber would also be required for the reference arm which would then need compensating. This, to me, seems like a significant drawback of this approach.

Reviewer #3 (Remarks to the Author):

I thank the authors for their replies to my comments and the changes included in the manuscript. I am satisfied with their responses and support the publication in Nature Communications

Reply to reviewer 1

We thank reviewer 1 for providing useful revision suggestions and we have edited the files accordingly (changes in the main text are highlighted).

- **Comment:** The range used should be included.
- **Reply:** We now more specifically state that the distance of the target object to the receiver and collimator is 4 meters.
- **Comment:** A discussion of the optical delay line should be included
- **Reply:** A new section in supplementary material is now added to discuss this topic.
- **Comment:** The dispersion should be mentioned. Although I agree with the authors' reply that dispersion in air over kilometre distances will minimally impact the results, kilometres of fiber would also be required for the reference arm which would then need compensating. This, to me, seems like a significant drawback of this approach.
- **Reply:** We now more specifically state in the main text that the dispersion of the probe and reference light needs to be properly compensated in order to reach maximal c-SFG conversion efficiency.